# Decoupled choice-driven and stimulus-related activity in parietal neurons may be misrepresented by choice probabilities

Adam Zaidel[1,2], Gregory C. DeAngelis [3] & Dora E. Angelaki[2]

Trial-by-trial correlations between neural responses and choices (choice probabilities) are often interpreted to reflect a causal contribution of neurons to task performance. However, choice probabilities may arise from top-down, rather than bottom-up, signals. We isolated distinct sensory and decision contributions to single-unit activity recorded from the dorsal medial superior temporal (MSTd) and ventral intraparietal (VIP) areas of monkeys during perception of self-motion. Superficially, neurons in both areas show similar tuning curves during task performance. However, tuning in MSTd neurons primarily reflects sensory inputs, whereas choice-related signals dominate tuning in VIP neurons. Importantly, the choice-related activity of VIP neurons is not predictable from their stimulus tuning, and these factors are often confounded in choice probability measurements. This finding was confirmed in a subset of neurons for which stimulus tuning was measured during passive fixation. Our findings reveal decoupled stimulus and choice signals in the VIP area, and challenge our understanding of choice signals in the brain.

---

[1] Gonda Multidisciplinary Brain Research Center, Bar Ilan University, Ramat Gan 52900, Israel. [2] Department of Neuroscience, Baylor College of Medicine, Houston, TX 77030, USA. [3] Department of Brain and Cognitive Sciences, Center for Visual Science, University of Rochester, Rochester, NY 14627, USA. Gregory C. DeAngelis and Dora E. Angelaki contributed equally to this work. Correspondence and requests for materials should be addressed to A.Z. (email: adam.zaidel@biu.ac.il)

The observation that activity of single sensory neurons in the brain can predict perceptual decisions, even before they are reported by the subject, has generated substantial and continued interest[1–3]. This result has been corroborated by widespread findings of significant choice probabilities (CPs; a metric that quantifies the relationship between neuronal activity and perceptual decisions across repeated presentation of a stimulus) in many cortical areas[1, 4–10] and recently, even subcortically[11].

However, the interpretation of CPs is complex[3]. According to a bottom-up (feedforward) interpretation, sensory responses are corrupted by noise, which propagates to downstream areas and influences choices, giving rise to significant CPs[1, 12]. According to this notion, CPs reflect a causal contribution of sensory neurons to decisions. However, top-down signals also modulate neuronal activity, even in early sensory areas[13–16], and these top-down signals may drive CPs. According to this notion, CPs could reflect high level factors such as context, attention, or other decision related influences[8, 17–21]. The reality, of course, may be that CPs reflect a combination of both bottom-up and top-down signals[22, 23], but the respective contributions of these influences are difficult to unravel.

An important difference between bottom-up and top-down origins of CPs involves the expected directionality of their influence on the firing rates (FRs) of sensory neurons. In a simple feedforward system, neuronal fluctuations should generally influence choices in a manner that is predictable from the neuron's tuning. For a neuron that prefers rightward motion stimuli, an above average response to a given stimulus should bias the animal's choice toward rightward in such a feedforward scheme. This produces values of CP > 0.5, by the usual convention in which CPs are computed, reflecting that choice-related modulations are in the same direction as the cell's tuning. By contrast, if CPs arise through top-down influences, choice-related modulations could be unrelated to a cell's tuning preference. This might result in a similar prevalence of CPs < 0.5, such that responses are weaker when the animal makes a choice in favor of the neuron's preferred stimulus. Although CPs < 0.5 are much less-frequently encountered in the literature, there is a potential confound in the way that CPs are generally computed that could bias CPs toward greater values, as described below.

We address the fundamental question of whether choice-related modulations are predictable from a neuron's tuning to task-relevant stimuli. Importantly, we show that CPs are a poor means to answer this question. This is because of a potential logical flaw: if top-down (choice related) signals are present in neuronal activity, then these choice signals influence the tuning curve from which a neuron's stimulus preference is determined (and used to compute CP). We show that this can create an artificial predisposition for CPs > 0.5. Consider two neurons: one that encodes only a sensory stimulus (e.g., a neuron with a rightward motion preference), and another that only receives feedback signals regarding perceptual decisions (e.g., rightward choices). Since, in a sensory discrimination task, choices are strongly coupled to stimulus values, both cells will show similar "tuning" when plotted as a function of the stimulus. For the neuron receiving top-down signals, the apparent stimulus preference results from choice signals, thus ensuring that CP > 0.5. This problem extends to neurons that carry both stimulus and choice signals, as long as the choice signals are large enough to override the sensory inputs and determine the cell's tuning preference. Thus, the fact that a neuron increases its firing rate for choices in favor of its "preferred stimulus" could result trivially from inadequate separation of sensory and choice signals in the traditional calculations of tuning curves and CPs. Hence, it is imperative to separate and quantify the concurrent influences of sensory and decision signals.

Here we performed such analyses on neurons recorded from the dorsal medial superior temporal (MSTd) and the ventral intraparietal (VIP) areas, two regions in which multisensory signals regarding the direction of self-motion (heading) have been well described[24–28]. Although CPs are significantly greater in VIP than MSTd[6, 10, 29], counter-intuitively, inactivation of VIP does not affect behavioral thresholds[30] (in contrast to MSTd[31]). This result may suggest that choice-related activity in VIP arises from top-down signals, rather than from a causal contribution of VIP to heading perception. We present a novel approach to dissociate sensory and choice signals, and thereby reveal a salient difference between these two areas. Our results demonstrate a predominance of choice signals in VIP (unlike MSTd), with a large unique contribution that is uncorrelated with the relevant sensory signals. These findings are consistent with mixed selectivity of distinct top-down decision, and bottom-up sensory, signals in VIP neurons. Furthermore, our results expose a major potential pitfall of traditional CP analyses that may call for reappraisal of some previous conclusions.

## Results

**Confounded heading and choice signals in tuning curves.** The activity of single neurons in areas VIP ($N = 307$) and MSTd ($N = 182$) was recorded from two monkeys performing a heading discrimination task. In addition to these primary data ("new" data set), data from two previous studies ("old" data set), comprising 106 VIP cells from Chen et al.[10] and 265 MSTd cells from Gu et al.[6], were also analyzed (see "Methods" section). Neurons were tested with either vestibular (inertial whole-body motion) or visual (optic flow) stimuli that corresponded to translations of the monkey along a straight path, with a Gaussian velocity profile (see "Methods" section for details). All headings were forward, but with slight deviations to the right or left of straight ahead. After each stimulus presentation, the monkeys were required to report whether their perceived heading was to the right or to the left of straight ahead (two-alternative forced choice task). Figure 1a, b presents the data from two example sessions, with interleaved vestibular (*blue*) and visual (*red*) stimuli, and simultaneously recorded single-unit activity from VIP neurons. Psychometric functions (*top row*) plot the proportion of rightward choices as a function of heading, with cumulative Gaussian fits providing a quantitative measure of behavioral performance.

Neuronal tuning curves (Fig. 1, *second row*, *blue* and *red curves*), which plot average firing rate (averaged over the stimulus period) as a function of heading, show that the example cells in Fig. 1a, b have rightward and leftward heading preferences, respectively. Although such tuning curves are generally assumed to reflect effects of the external stimulus, it should be kept in mind that the ratio of choices in favor of preferred and null headings will covary with the stimulus. For stimuli around straight ahead, where discrimination is difficult, decisions will comprise a mixture of rightward and leftward choices. By contrast, leftward or rightward choices will predominate for large negative or positive headings, respectively. Thus, for neurons that receive top-down signals related to choice (or other cognitive states that may covary with the stimulus), the tuning curve generally reflects some (unknown) combination of stimulus and choice signals, and additional analyses are needed in order to tease apart these contributions. This is clearly a potential concern for tuning curves measured during discrimination tasks, and it is possible that choice signals may also contaminate tuning curves measured during passive viewing in trained animals (see "Discussion" section).

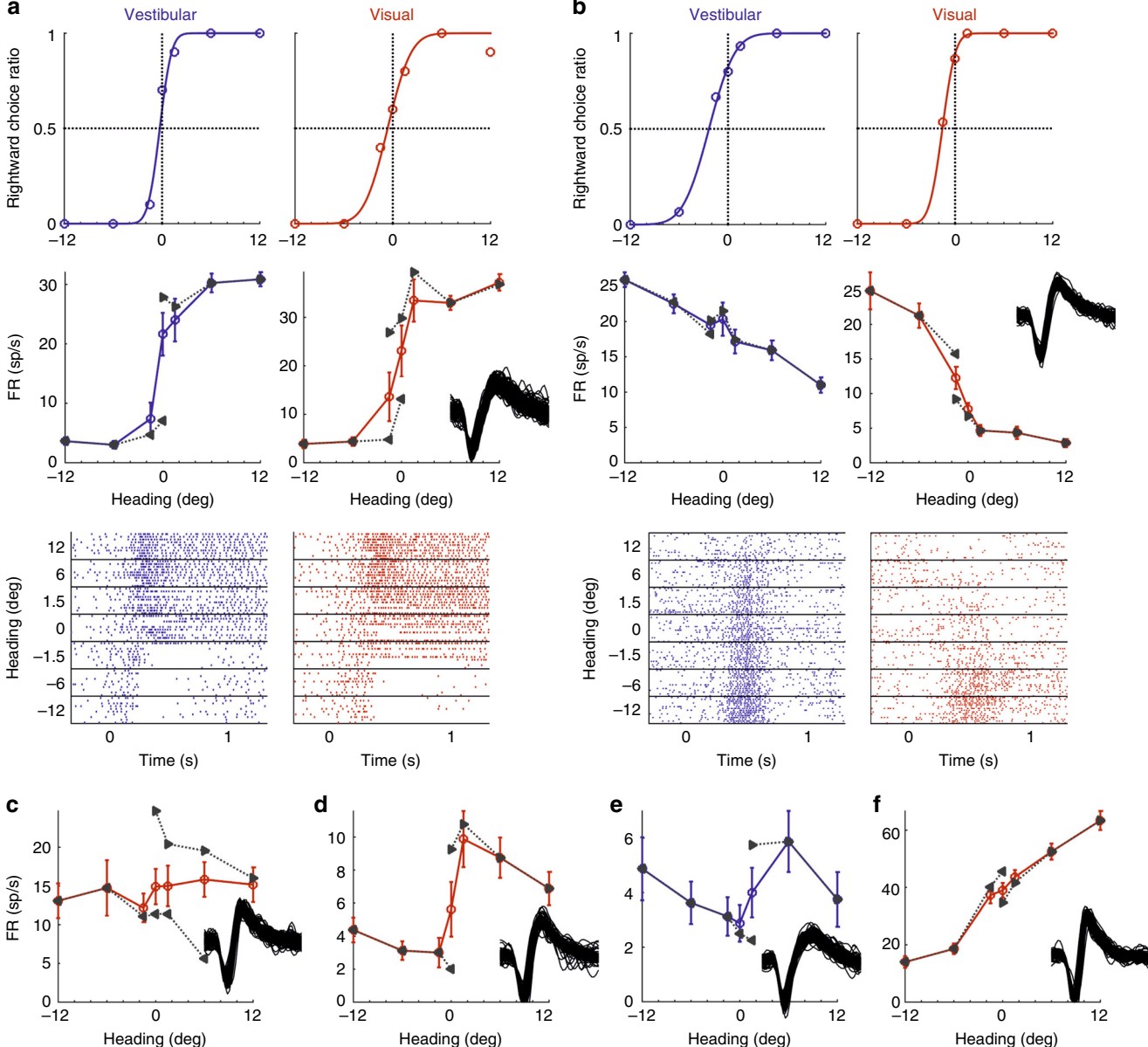

**Fig. 1** Example single-unit recordings from awake, behaving monkeys during heading discrimination. **a**, **b** The behavioral data from two example sessions, along with simultaneous single-unit recordings from VIP. *Top row*: psychometric curves represent the ratio of rightward choices as a function of heading, based on visual (*red*) or vestibular (*blue*) cues. The data (*circles*) were fitted with cumulative Gaussian functions (*solid curves*). *Second row*: average firing rates (FRs) are presented as a function of heading. *Leftward* and *rightward* pointing *gray triangles* mark the average FRs for leftward and rightward choices, respectively. For each unit, one hundred (randomly selected) overlaid spikes are presented (*insets*). *Third row*: raster plots depict the spikes for each trial, as a function of time. Stimulus motion, having a Gaussian velocity profile, spanned the epoch from 0 to 1 s. *Horizontal black lines* in the raster plots separate different headings. **c–f** Average FRs as a function of heading are presented for four additional VIP units (3 units tested with the visual condition in *red*, and 1 unit tested with the vestibular condition in *blue*) from four separate sessions. All *error bars* denote SEM

In order to demonstrate this point, we calculated average FRs separately for trials followed by leftward vs. rightward choices (leftward and rightward pointing *gray triangles*, respectively, Fig. 1). The resulting choice-conditioned tuning curves show heading tuning when choice is held constant (only headings with ≥ 3 repeats of the conditioned choice were included). An effect of choice on FR can be discerned in Fig. 1a: the rightward-choice-conditioned responses are greater than the leftward-choice-conditioned responses (see the headings around 0° for both the visual and vestibular cues). Figure 1b also shows a choice effect for the visual cue; however, the vestibular cue seems to have less of a choice effect. Critically, for the examples in Fig. 1a, b (visual), the choice effect is in the same direction as the heading tuning – namely, choices to the side of the cell's preferred heading correspond with increased FRs.

However, this alignment of heading and choice signals is frequently not observed in VIP, as demonstrated by the example neurons in Fig. 1c–f. Figure 1c displays a cell with leftward heading tuning (both choice-conditioned curves have negative slopes) but a strong rightward choice preference (the rightward-choice-conditioned curve is well above the leftward-choice-conditioned curve). As seen in this example, heading and choice signals can cancel each other, resulting in an overall tuning curve (red) that is roughly flat. The example

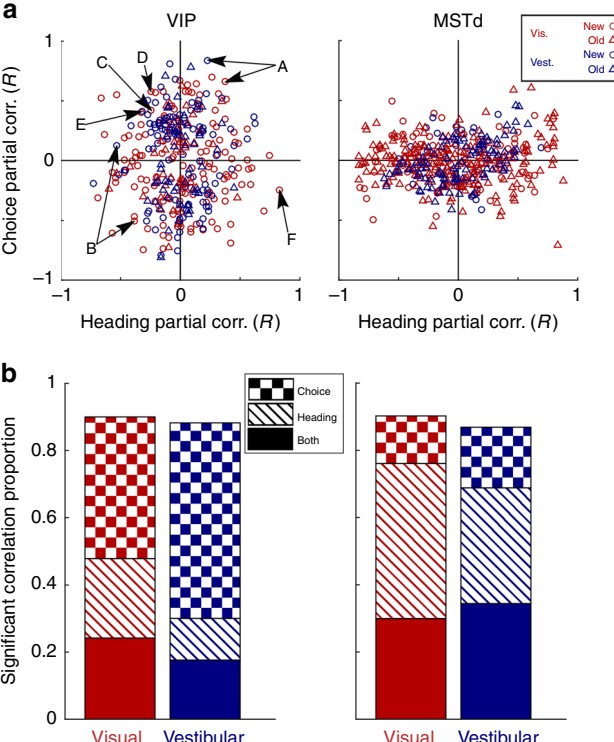

**Fig. 2** Heading and choice partial correlations. Partial correlations were calculated between neuronal firing rates and stimulus headings (given the actual choices made by the monkey) or choices (given the stimulus headings). Only cells that showed a significant main effect of heading or choice were included ($p < 0.05$, two-way ANOVA). **a** Partial correlations are presented as scatterplots for VIP (*left*) and MSTd (*right*). *Red* and *blue* symbols represent the data from the visual and vestibular conditions, respectively, and circle and triangle symbols represent the new and old data sets, respectively. *Arrows* indicate the data points corresponding to the VIP tuning curves presented in Fig. 1 (letters A–F for the respective Fig. 1 subplots). **b** The proportion of neurons (same data as in **a**) with significant partial correlations for choice-only, heading-only or both heading and choice. In VIP, neurons with significant choice partial correlations are prevalent; while MSTd shows a greater proportion of significant heading partial correlations. In **b**, the new and old data were pooled for each brain area (results were similar for each data set individually). For VIP new (old) data, $N = 140$ (50) and 95 (58) for visual and vestibular conditions, respectively. For the MSTd new (old) data, $N = 80$ (217) and 25 (97) for visual and vestibular conditions, respectively. See also Supplementary Fig. 1 for a simulation and Supplementary Figs. 2–4 for further analyses of these data

neurons in Fig. 1d, e also display leftward heading preferences and rightward choice preferences. However, the choice preference dominates the response and dictates the overall shape of the tuning curves, which, when viewed on their own, would (incorrectly) suggest that these neurons preferred rightward headings. As described above, such neurons are problematic for the traditional calculation of CPs since the cell's apparent heading preference is caused by choice signals, thus ensuring that CP > 0.5. Finally, Fig. 1f shows a neuron with rightward heading tuning, and a modest preference for leftward choices that does not have much impact on the overall shape of the tuning curve. These examples clearly expose the need for analysis methods that can identify the stimulus- vs. choice-related contributions to neuronal activity.

**Stimulus and choice partial correlations**. To estimate the distinct influences of heading and choice on neuronal activity, we performed multiple regression analyses (see "Methods" section) and calculated the effect of one parameter (heading or choice) on response, given the other, using partial correlation analysis. This analysis provided two measures: $R$(FR, heading|choice), the partial correlation between FR and heading, given the monkeys' choices (i.e., 'heading partial correlation'); and $R$(FR, choice|heading), the partial correlation between FR and choice, given the stimulus headings (i.e., 'choice partial correlation'). Each measure thus represents the distinct contribution of that parameter to the neuronal response. By convention, positive (negative) heading partial correlations reflect an increase in FR for rightward (leftward) headings; and positive (negative) choice partial correlations reflect an increase in FR for rightward (leftward) choices. As described below, heading and choice partial correlations need not have the same sign. Validation of this method with simulations is presented in Supplementary Note 1 and Supplementary Fig. 1; also, additional analysis that does not assume linear fits is described below.

Although heading tuning curves for MSTd and VIP neurons generally look similar[27–29], we find that there are substantial differences between areas in the relative contributions of heading and choice signals to tuning curves, as revealed by partial correlation analysis (Fig. 2). In VIP, choice partial correlations are prevalent, as observed by the vertical spread of the partial correlation data (Fig. 2a, *left*) and the greater proportion of cells with significant choice, as compared to heading, partial correlations (Fig. 2b, *left*). By contrast, MSTd shows dominance of heading partial correlations, as observed by the horizontal spread of the partial correlation data (Fig. 2a, *right*) and the greater proportion of cells with significant heading partial correlations (Fig. 2b, *right*). Accordingly, type-II regression analysis of the relationship between choice and heading partial correlations (Supplementary Fig. 2) indicates greater spread along the vertical (choice) and horizontal (heading) axes for VIP and MSTd, respectively.

Importantly, these data do not reveal significant positive slopes in the relationship between heading and choice partial correlations (Pearson correlation $p > 0.05$) for seven out of the eight combinations of area (VIP or MSTd), modality (visual or vestibular), and data set (new or old), with one exception discussed further below (Pearson correlation $p = 1.7 \times 10^{-8}$ for MSTd, vestibular condition, old data set; $N = 140$ and 95 for visual and vestibular conditions, respectively, in the new VIP data set, and 50 and 58, respectively, in the old VIP data set; $N = 80$ and 25 for visual and vestibular conditions, respectively, in the new MSTd data set, and 217 and 97, respectively, in the old MSTd data set). Data points falling in the top-right and bottom-left quadrants of Fig. 2a correspond to neurons for which the effect of choice is consistent with the effect of the stimulus, as expected for a simple feedforward scheme in which neurons provide evidence in favor of their preferred stimulus. In contrast, data points falling in the top-left and bottom-right quadrants indicate neurons for which the effect of choice opposes the effect of the heading stimulus; for example, FR increases for rightward heading stimuli but decreases for rightward choices, as demonstrated by the locations of the data points in Fig. 2a that correspond to the neurons from Fig. 1c–f. The roughly equal distribution of data across the four quadrants suggests that at least some of the choice effects in VIP and MSTd have a top-down origin. As shown in the next section, this finding is partly in agreement with the observation of CP values < 0.5, but this phenomenon is much more widespread than appreciated by computing CPs. This is because of the ambiguity involved in using the tuning curve to define the preferred stimulus direction when the tuning curve is also influenced by choice (Fig. 1).

The analysis of Fig. 2 relies on linear partial correlations of FR with heading and choice. Specifically, it assumes a relationship of

the form $FR = \beta_{heading}\cdot heading + \beta_{choice}\cdot choice + C$, where heading and choice are the trial-by-trial values of the stimulus and choice (coded as $+1$ for rightward choices and $-1$ for leftward choices), $\beta_{heading}$ and $\beta_{choice}$ are the coefficients determined by regression, and $C$ is a constant. Although most heading tuning curves are fairly linear over the narrow range tested in the task[6], a more complex model, which allows for non-linear heading tuning and interaction effects between heading and choice, may provide a better fit. Moreover, variance related to heading might be misattributed to choice if the model does not adequately fit heading tuning.

Thus, we also calculated partial correlations using a more complex regression model (see Supplementary Note 2), and the results remained very similar (Supplementary Fig. 3A). Although the more complex model improved the fits significantly for a substantial fraction of neurons (Supplementary Fig. 3B), the variance explained by the linear heading and choice terms was much greater than that explained by the non-linear heading term or the heading-choice interaction term (Supplementary Fig. 3C). Importantly, the linear terms in the more complex model again clearly demonstrate choice-dominance in VIP and heading-dominance in MSTd (Supplementary Fig. 3C).

**Choice probabilities are biased**. To understand the relationship between the partial correlation analysis described above and the traditional computation of CPs, we plotted CPs as a function of both choice and heading partial correlations (Fig. 3, *green-orange* heat maps). Although the distributions of partial correlations vary somewhat across the two areas and the two stimulus modalities, the relationship between CPs and the partial correlations is similar. This visualization also reveals an important difference between CPs and choice partial correlations: there are substantial portions of the top-left and bottom-right quadrants in which CPs are $> 0.5$ (*green shades*), yet the heading and choice partial correlations have opposite signs. The net result is that CPs are unbalanced across the plane and biased toward values $> 0.5$.

The primary reason for the unbalanced distribution of CPs can be seen by first examining data that lie near the vertical axis of the heat maps in Fig. 3. Points near the vertical axis correspond to neurons with large choice signals and little or no heading signals. For these cells, it is the choice signals that dominate the tuning curve and determine the "preferred heading". Since CPs are computed by sorting responses into two groups (corresponding to preferred and null choices) based on the tuning curve, neurons with dominant choice signals will trivially have high CPs. For such neurons, a large CP $> 0.5$ might seem to imply that choice signals are consistent with stimulus tuning (as expected in a feedforward scheme), but this is an artifact of not properly separating the contributions of choice and heading to the tuning curve.

By contrast, around the horizontal axes in Fig. 3, CPs tend to be close to 0.5 (i.e., chance, no effect of choice). Furthermore, there is a portion of the top-left and bottom-right quadrants, in which CPs are consistently $< 0.5$ (*orange colors*). These areas correspond to neurons with large heading partial correlations and small choice partial correlations of opposite sign, such that the tuning curve is dominated by heading signals. In these cases, CPs correctly reflect the fact that heading and choice signals have opposite effects on FRs. Critically, this corresponds to a small region of the partial correlation plane, resulting in the finding that CPs, overall, are artifactually biased toward values $> 0.5$.

These considerations imply that CPs will tend to be $> 0.5$ on average even when there is no systematic relationship between heading and choice partial correlations, as suggested by the data of Fig. 2 (see also Supplementary Figs. 2, 3). Notably, this happens even in simulations for which heading and choice signals are

completely uncorrelated (Supplementary Fig. 1C): the resulting CPs were clearly biased toward values $> 0.5$ (mean CP $= 0.63$; significantly different from 0.5; t-test, $p < 10^{-12}$, $N = 300$ simulated cells). Hence, when choice signals are large enough to dictate tuning preferences, CPs will not provide an accurate metric of the relationship between stimulus and choice influences, and may suggest greater consistency with a feedforward model than is warranted. As explored further in the Discussion, this bias in CPs is likely to be smaller in areas for which top-down signals are weaker.

As revealed by Fig. 3, the finding of greater CPs in VIP than MSTd does not reflect a stronger coupling between stimulus and choice signals; rather, it primarily reflects the broader distribution of VIP data along the vertical (choice) axis. Thus, an analysis that correctly dissociates response components related to stimulus and choice represents a vital approach towards understanding the origins of choice-related activity. For the partial correlation analysis presented here, any response variation that could be explained by both heading and choice (which are themselves correlated) was removed. Plotting the beta coefficients from the multiple regression analysis, rather than partial correlations, demonstrates very similar results (Supplementary Fig. 4), but does not discard the common components of response variation. However, this analysis has other limitations due to parameter normalization (see Supplementary Note 3 for further details).

**Dynamics of partial correlations**. Thus far, partial correlations have been computed based on mean FRs measured over the length of each trial, providing a single pair of heading and choice partial correlation values per neuron. But there are inherent response dynamics that are at least partly due to the stimulus motion profile. Thus, we explored the time-course of heading and choice signals by computing partial correlations as a function of time across the trial (see "Methods" section for details). For this analysis, we inverted the signs of the partial correlations for neurons with negative heading partial correlations. Namely, if a neuron's overall heading partial correlation was negative, then we inverted the sign of the time courses of heading and choice partial correlations for that neuron. Thus, each neuron's heading partial correlations over time are mainly positive (but could have negative values for some time periods), and positive (negative) choice partial correlation values reflect increased FRs for choices toward the preferred (non-preferred) heading. This was done separately for data from the visual and vestibular conditions.

Figure 4a shows the resulting average partial correlations as a function of time. Here we see that the average choice partial correlation curves remain close to zero across the stimulus interval. This does not indicate weak or absent choice partial correlations. To the contrary, strong choice components in VIP are exposed when plotting the average $R^2$ values (Fig. 4b). Rather, the result of Fig. 4a indicates that the choice partial correlations cancel out because they are largely unrelated to the heading partial correlations across time.

The time-course plots also reveal that the heading partial correlations rise and fall in a bell-shaped manner for both VIP and MSTd, roughly similar to the stimulus velocity profile (gray curves). For VIP, the choice responses rise during stimulus presentation and remain elevated throughout the remainder of the trial epoch (Fig. 4b, *left*). For MSTd, choice-related activity is much weaker, but also rises during the stimulus presentation and is maintained through the end of the trial (Fig. 4b, *right*). Importantly, the same partial correlation analysis was applied separately to the instantaneous FR at each time step, using the choice and heading parameters for each trial (constant values per trial, not dependent on time). Thus the shapes of the time courses

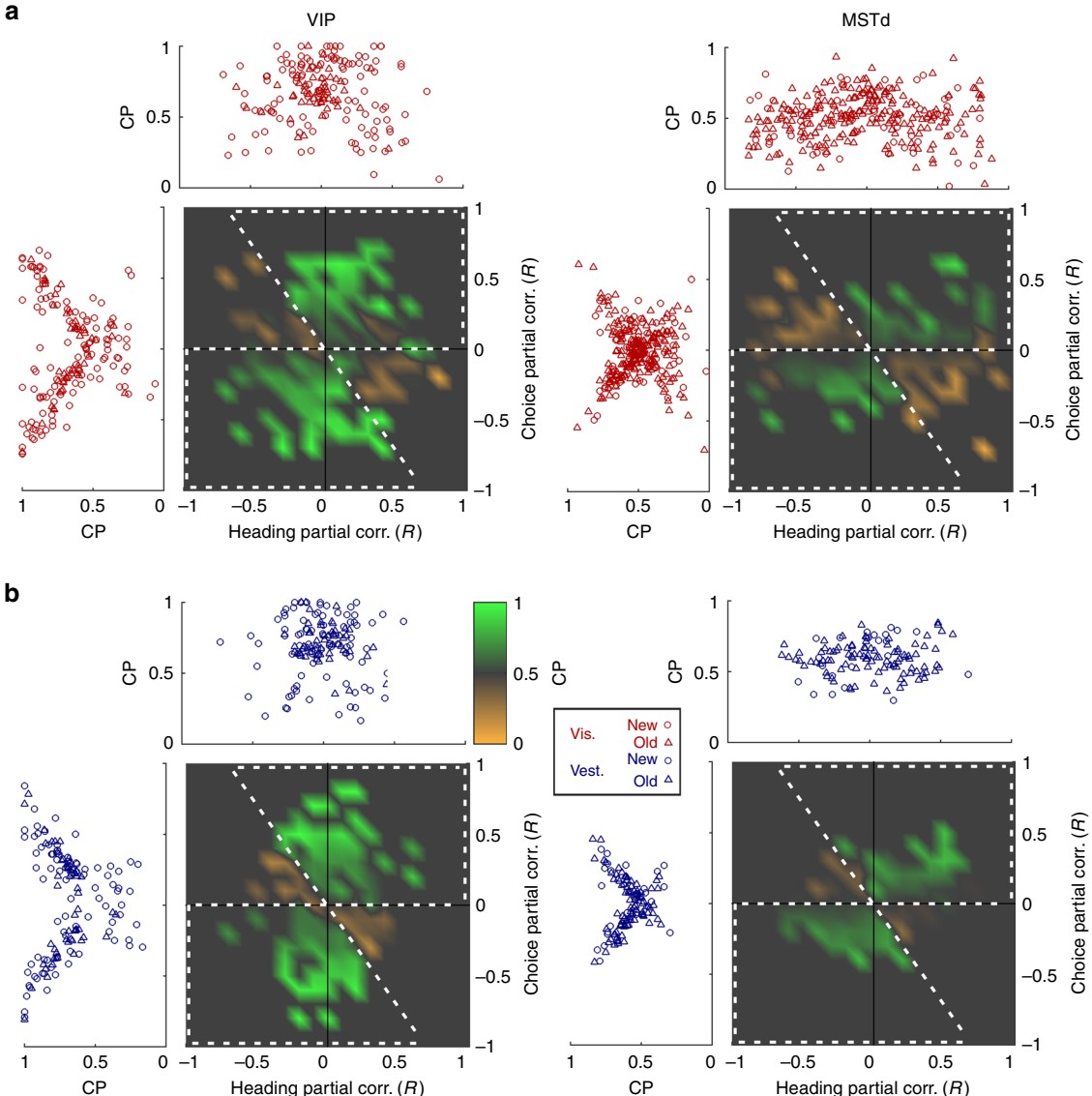

**Fig. 3** Relationship between heading and choice partial correlations and choice probabilities. Average choice probability (CP; *green-orange* heat map) is presented in the plane of partial correlations for the visual (**a**) and vestibular (**b**) data from VIP (*left* column) and MSTd (*right* column). The background *gray color* represents CP = 0.5; shades of *green* and *orange* represent CP > 0.5 and CP < 0.5, respectively. *White dashed lines* (drawn by eye, identical for all four subplots) outline the regions characterized predominantly by high CPs (CP > 0.5). Marginal plots show CP values plotted against heading and choice partial correlations. Cell samples and *N* values are the same as in Fig. 2. Circle and triangle symbols mark the new and the old data sets, respectively. See also Supplementary Fig. 1D for CP simulation

observed in Fig. 4 are empirical, and are, in no way, imposed by the analysis itself.

To compare the time courses of choice partial correlations between VIP and MSTd quantitatively, squared choice partial correlations ($R^2$; Fig. 4b) were normalized, such that 0 and 1 represent their minimum and maximum values, respectively, and overlaid (Supplementary Fig. 5). A two-way ANOVA was applied to data from the first half of stimulus presentation, with data grouped according to: a) condition (4 possible conditions of 2 stimulus modalities, visual and vestibular, and two brain areas, VIP and MSTd; $N = 190$ and 153 for VIP visual and vestibular conditions, respectively, and 297 and 122 for MSTd visual and vestibular conditions, respectively), and b) normalized time (one of 5 possible values: 0.1, 0.2, 0.3, 0.4 or 0.5). This analysis revealed a significant main effect of condition ($p = 2.8 \times 10^{-6}$) and, trivially, also a main effect of time ($p = 2.8 \times 10^{-65}$). Post hoc (Tukey-Kramer) tests revealed that choice signals in VIP neurons

rose significantly earlier for vestibular stimuli than for visual stimuli ($p = 2.1 \times 10^{-5}$; Supplementary Fig. 5). These VIP choice signals for vestibular stimuli also rose significantly earlier than choice signals in MSTd for visual stimuli ($p = 1.6 \times 10^{-5}$), but not in MSTd for vestibular stimuli ($p = 0.18$). For the three other conditions (VIP visual, MSTd visual, and MSTd vestibular) choice signals were not significantly different from one another ($p > 0.1$), although the same trend was seen for choice signals to rise more quickly in the vestibular than visual condition in MSTd. These results may provide support for the proposal that acceleration signals are primarily used for vestibular motion perception, whereas visual heading perception may rely primarily on velocity information that peaks slightly later[32].

**Congruent and opposite cells**. Overall, we found no clear relationship between heading and choice partial correlations in VIP and MSTd (Figs. 2–4). This contrasts with the bottom-up

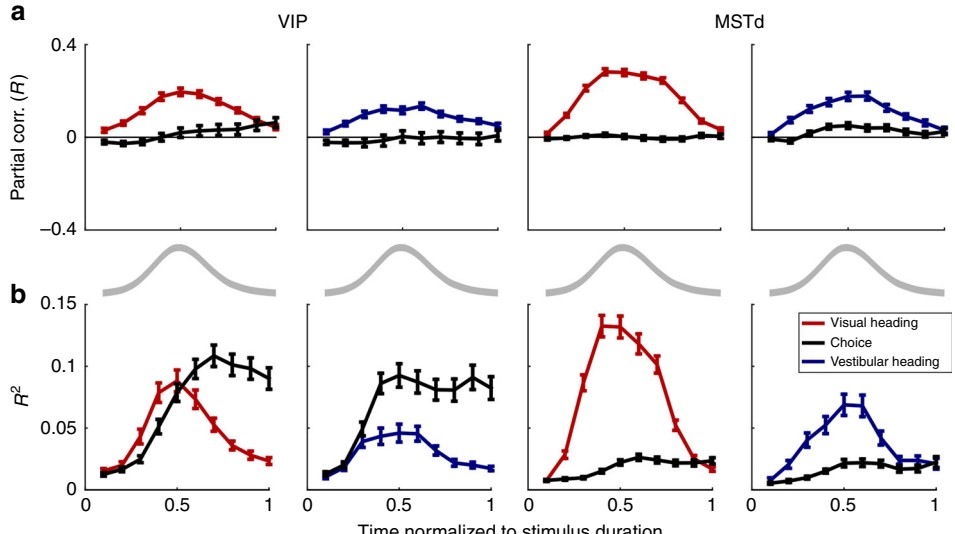

**Fig. 4** Partial correlations as a function of time. Partial correlation data are presented for areas VIP (*left* two columns) and MSTd (*right* two columns). Heading partial correlation data are presented in *red* for the visual condition and in *blue* for the vestibular condition. The respective choice partial correlations are presented in *black*. **a** The average partial correlations were aligned to each cell's heading preference by inverting the signs of both the heading and choice partial correlations when the overall heading partial correlation was negative (done separately for visual and vestibular conditions). On average, the choice partial correlations across cells cancel out, indicating no consistent alignment to heading preferences. **b** Squared partial correlations ($R^2$) are presented, such that positive and negative contributions from different neurons did not cancel. The *gray lines* below **a** depict the stimulus velocity profile. Cell selection and N values are the same as in Figs. 2 and 3. The old and new data were pooled in this plot, and the horizontal axis shows time normalized to stimulus duration, which was different for the new and old data sets. See also Supplementary Fig. 5 for a more direct comparison of the choice partial correlations across areas and stimulus modalities ($R^2$)

explanation of CPs, for which we would generally expect heading and choice partial correlations to have the same sign. Importantly, this prediction of a bottom-up scheme assumes that population responses are decoded such that each neuron provides evidence in favor of its preferred stimulus. A complication then arises for multisensory neurons, known as 'opposite' cells[6, 26, 28], that have different heading preferences for visual and vestibular cues. Opposite cells could potentially be decoded to provide evidence in favor of either their visual or vestibular heading preference; in contrast, this issue does not arise for 'congruent' cells that have matched visual and vestibular preferences.

Recent studies have suggested that MSTd neurons, including both congruent and opposite cells, may be decoded according to the vestibular heading preference of each cell[6, 33, 34], and this arrangement has been shown to qualitatively predict the pattern of CPs observed for visual heading discrimination[33]. Importantly, such a decoding strategy would predict that heading and choice partial correlations would have different signs for opposite cells in the visual condition, even in a bottom-up scenario. Thus, it is possible that the result of Fig. 2a (particularly for the visual condition) reflects different relationships for congruent and opposite cells. Hence, we examined heading and choice partial correlations separately for these two groups of neurons.

Given that heading and choice signals are mixed in tuning curves, we classified congruent and opposite cells based on their heading partial correlations. Specifically, cells were classified as congruent (or opposite) if their visual and vestibular heading partial correlations had the same (or opposite) signs (Fig. 5a). This ensures that the classification of cells is not confounded by choice signals. In addition, cells were labeled as significantly congruent or opposite if both the visual and vestibular heading partial correlations were significant ($p < 0.05$). For MSTd, 61 and 33 cells were classified as congruent and opposite, respectively, with 42 and 16 of these being significant. For VIP, 64 and 30 cells were classified as congruent and opposite, respectively, but only

12 congruent and 1 opposite cell were significant (Fig. 5a). This reflects the weaker contribution of heading signals to VIP responses, as compared to MSTd.

Interestingly, we find that congruent cells in MSTd show a positive relationship between heading and choice partial correlations for both the visual (*red*) and vestibular (*blue*) conditions (Fig. 5b, third column; $p = 0.0004$ and $p = 4 \times 10^{-8}$, respectively; Pearson correlation between heading and choice partial correlations; $N = 61$). This indicates that choice-related signals in MSTd congruent cells are largely predictable from stimulus tuning. Critically, for opposite cells in MSTd, we find a significant negative relationship between heading and choice partial correlations in the visual condition (Fig. 5b, *red* plot, *rightmost* column; Pearson correlation $p = 0.02$, $N = 33$). Thus, results for the visual condition in MSTd are consistent with the idea that MSTd neurons may be decoded according to their vestibular heading preferences[6, 33].

This theory also predicts a positive relationship between heading and choice partial correlations for MSTd opposite cells in the vestibular condition. While a positive trend was seen (*rightmost* column, Fig. 5b, *blue*), this was not significant (Pearson correlation $p = 0.47$, $N = 33$). It should be noted, however, that there were fewer opposite vs. congruent cells, and that vestibular responses in the new MSTd data set were less robust (see Methods). Thus, the vestibular condition results for MSTd opposite cells are inconclusive on their own. However, visual and vestibular choice signals were positively correlated for both congruent and opposite cells (Pearson correlations $p = 2 \times 10^{-6}$ and $p = 4 \times 10^{-4}$, with $N = 61$ and 33, respectively; Fig. 5c), and vestibular and visual heading preferences of 'opposite' cells are opposite by definition. Thus, taken together with the negative correlation observed between heading and choice signals for opposite cells in the visual condition (described above), these results suggest systematic relationships between MSTd heading and choice signals that could arise from selective

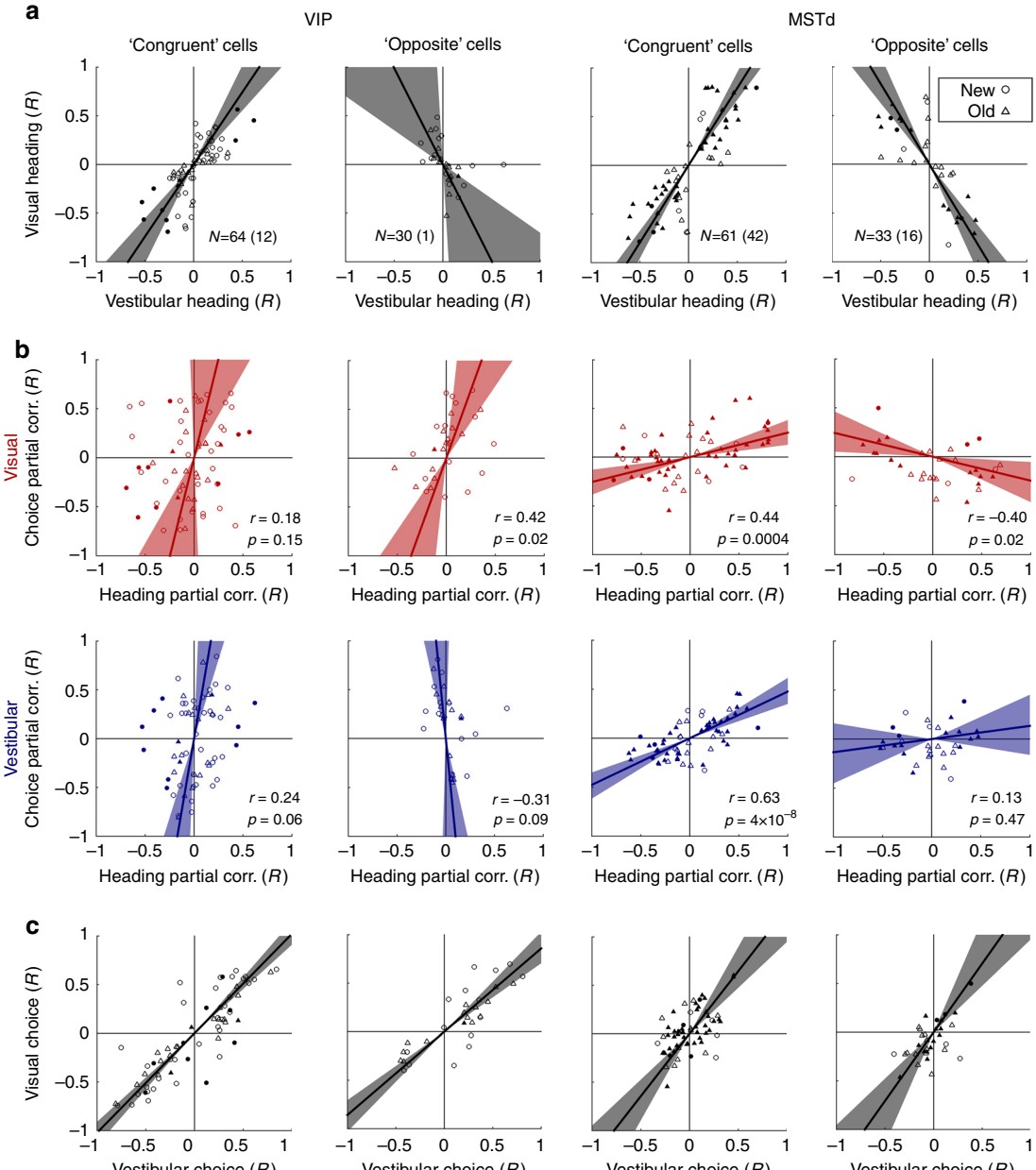

**Fig. 5** Partial correlation results for congruent and opposite cells. **a** Visual and vestibular heading partial correlations are presented for neurons classified as 'congruent' or 'opposite' based on whether the visual and vestibular partial correlations for heading had the same or opposite signs, respectively. Significantly congruent or opposite cells ($p < 0.05$ for both the visual and vestibular heading partial correlations) are marked by filled data points. Circle and triangle symbols mark the new and the old data sets, respectively. $N$ values represent the number of cells (the number of significantly congruent or opposite cells in parentheses). Only cells which showed a significant main effect of heading or choice ($p < 0.05$, two-way ANOVA) for both visual and vestibular conditions were included. **b** Heading and choice partial correlations are presented for the same cells. Red and blue symbols represent the data for the visual and vestibular conditions, respectively. **c** Choice partial correlations measured during the visual and vestibular conditions are compared for the same cells. Solid lines and the shaded regions represent type-II regressions with their 95% confidence intervals. $r$ and $p$-values for the regressions in panel **b** are presented on the respective plots. For all the regressions in black (**a**, **c**) $p < 0.001$ (except for the second plot in **a**, for which $p = 0.02$)

decoding according to vestibular preferences. Moreover, this congruency dependence may explain the overall lack of correlation between heading and choice partial correlations for the visual condition in MSTd (Fig. 2a, right).

Might the same explanation, in terms of congruent and opposite cells, account for the apparent independence of heading and choice signals in VIP? We found that this is not the case. Although significant congruent and opposite cells were rarer in VIP than MSTd, we found a positive slope between heading and choice partial correlations for VIP opposite cells in the visual

condition (Fig. 5b, red, second column; Pearson correlation $p = 0.02$, $N = 30$). This differs clearly from MSTd (which had a significant negative slope; Pearson correlation $p = 0.02$, $N = 33$), and is not consistent with decoding congruent and opposite cells according to their vestibular preferences. Rather, both cell types have similar results for VIP (Fig. 5b; note the largely overlapping confidence intervals for the visual condition in VIP, red), with steeply sloped relationships between heading and choice partial correlations that reflect the predominance of choice signals in VIP. Therefore, for our VIP data, varied congruency of visual and

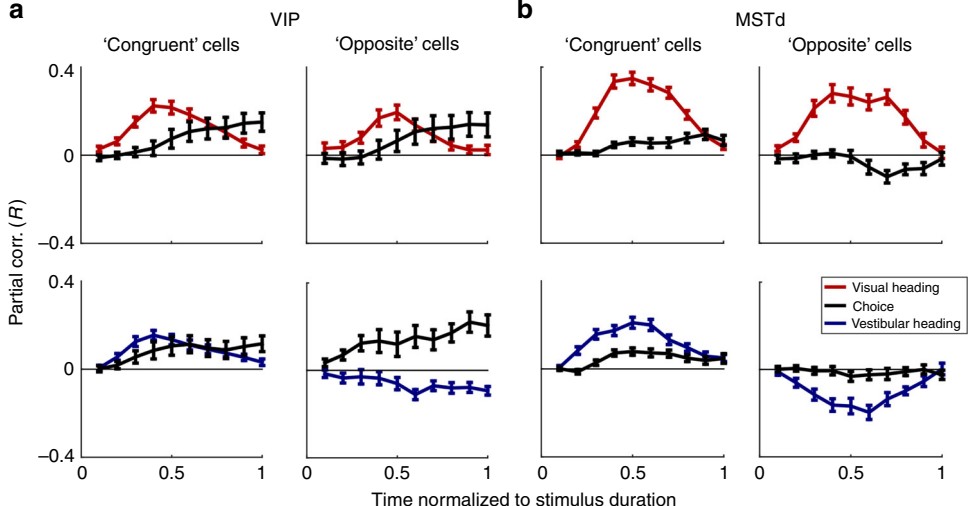

**Fig. 6** Partial correlation time-course for congruent and opposite cells. Average partial correlation (*R*) values are presented for congruent and opposite cells in VIP (**a**) and MSTd (**b**). Here, the data were aligned to each cell's visual heading preference by inverting the partial correlation signs such that the visual heading partial correlation was always positive. This was done together for visual and vestibular heading and choice partial correlations, such that positive choice (or positive vestibular heading) *R*-values indicate responses with the same sign as the visual heading partial correlation. Cell selection and *N* values are the same as in Fig. 5. The old and new data were pooled in this plot, and the times along the horizontal axis are normalized to stimulus duration. All *error bars* denote SEM

vestibular heading signals cannot explain the apparent overall independence of the unique choice and heading signals.

The time-course of partial correlations for congruent and opposite cells offers further insight into the relationship between heading and choice signals in VIP and MSTd (Fig. 6). For this analysis, the signs of the visual and vestibular heading and choice partial correlations (as a function of time) were all inverted if the overall visual heading partial correlation was negative. Note that here, unlike Fig. 4, the signs of the vestibular heading and choice partial correlations were inverted according to the sign of the visual heading partial correlation, to allow comparisons across cues.

For VIP in the visual condition, we see that both congruent and opposite cells have choice effects that are aligned to the same side as (i.e., with the same polarity as) the visual heading preference (Fig. 6a, *top*). Interestingly, VIP choice signals in the vestibular condition also have the same polarity as visual heading signals, for both congruent and opposite cells (Fig. 6a, *bottom*). Strikingly, the reverse pattern was seen for MSTd—choice signals, albeit weaker, generally have the same polarity as vestibular heading signals, such that the choice partial correlations for opposite cells have a polarity opposite to the visual heading signals (Fig. 6b, *right*). As described above (Fig. 4), this finding seems to be in line with a systematic relationship between MSTd heading and choice signals that could arise from selective decoding according to vestibular preferences.

**Heading tuning during passive fixation.** The results presented above hinge upon the ability of the partial correlation analysis to tease apart response components associated with stimulus and choice from data measured during the discrimination task. If a measurement of pure stimulus tuning were available, one would expect that pure stimulus tuning would be related to heading partial correlations, but not choice partial correlations. Thus, for a subset of neurons, we compared partial correlations with heading preferences measured during passive fixation. While we cannot firmly rule out any contribution of choice signals to heading tuning measured during these blocks of fixation trials

(see Discussion), these data provide additional leverage to evaluate the coupling of stimulus and choice signals.

For neurons in the old data set, global heading tuning in the horizontal plane (8 azimuths, 45° apart) was tested before running the heading discrimination task. In these separate blocks of trials, monkeys were simply rewarded for maintaining fixation—no choice was required. For the visual and vestibular conditions separately, we divided neurons into two groups based on whether their global heading preference was rightward (0° < preferred azimuth < 180°) or leftward (−180° < preferred azimuth < 0°). We then plotted distributions of heading and choice partial correlations separately for each direction of heading preference, brain area, and stimulus modality (Fig. 7). All cells with both global heading tuning data during fixation and heading discrimination data were included (*N* = 93 for VIP, one of which had only visual data, and *N* = 265 for MSTd).

The results reveal a clear relationship between heading preferences measured during passive fixation and heading partial correlations measured during the discrimination task, as seen by the shifts between distributions of heading partial correlations corresponding to rightward and leftward global heading preferences (Fig. 7a). The difference in heading partial correlations between rightward and leftward heading preferences was significant for all 4 combinations of brain area and sensory modality (*p*-values of Wilcoxon rank sum tests are presented in the respective plots).

By contrast, choice partial correlations are generally not related to global heading preferences measured during fixation (Fig. 7b). No significant difference in choice partial correlation is seen between groups of neurons that prefer rightward and leftward headings for VIP, visual and vestibular (Fig. 7b, *left*), and MSTd visual data (Fig. 7b, *top right*); with the only exception being the vestibular condition for MSTd (Fig. 7b, *bottom right*). This latter result is in line with our finding of a significant positive relationship between heading and choice partial correlations for the old MSTd data set in the vestibular condition (Supplementary Fig. 2). Overall, the pattern of results in Fig. 7b is consistent with the pattern seen in Fig. 2a, thus providing additional support for our main findings and our partial correlation analysis approach.

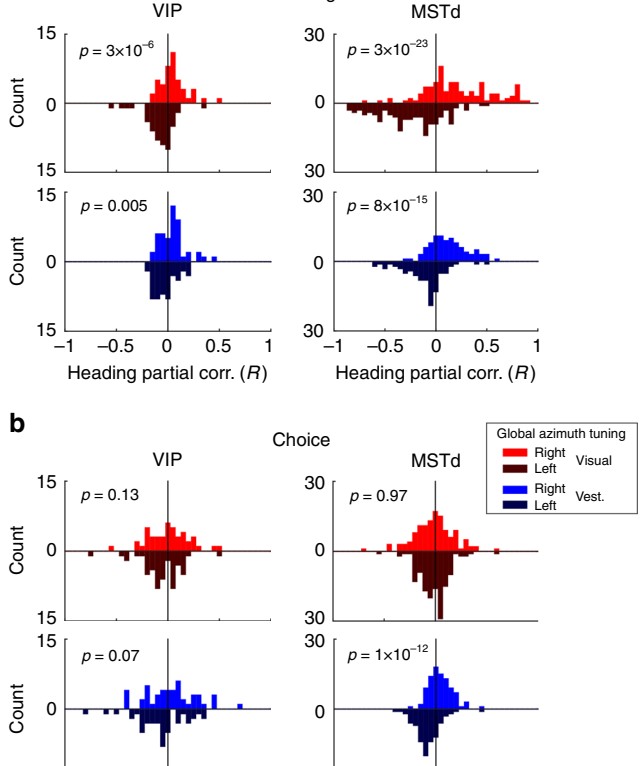

**Fig. 7** Heading, but not choice, partial correlations are predicted by heading preferences measured during fixation. (**a**) Heading and **b** choice partial correlation histograms are presented for the data with rightward and leftward global heading tuning preferences (*lighter-colored* upward histograms, and *darker-colored* downward histograms, respectively). The data are presented separately for VIP and MSTd (*left* and *right* columns, respectively) and visual and vestibular conditions (*red* and *blue* shades, respectively). *p*-values (Wilcoxon rank sum tests) are presented on the plots. N = 93 for VIP visual (49 classified 'Left' and 44 classified 'Right'), N = 92 for VIP vestibular (45 classified 'Left' and 47 classified 'Right'), N = 265 for MSTd visual (141 classified 'Left' and 124 classified 'Right'), and N = 265 for MSTd vestibular (124 classified 'Left' and 141 classified 'Right')

## Discussion

We investigated the contributions of distinct sensory and choice signals to neuronal activity in monkey areas MSTd and VIP. Using a novel approach to isolate these signals, we found that, although neuronal tuning curves in MSTd and VIP often look similar, responses in these areas reflect quite different compositions of sensory and choice signals. While MSTd responses are dominated by sensory (heading) signals, choice-related signals are most prevalent in VIP. Importantly, the specificity of the unique choice signals in VIP was generally not predictable from a neuron's heading tuning. Rather, choice signals make a distinct contribution to VIP responses, and sometimes overwhelm the stimulus-related components of neuronal activity. Therefore, these choice signals likely reflect the action of top-down feedback, as opposed to bottom-up effects of random fluctuations in response.

Our findings have important implications for the measurement and interpretation of CPs. CPs are computed by sorting neural responses into two choice groups—"preferred" and "null" choices—where these groups are defined according to each cell's stimulus preference. Thus, CP > 0.5 is generally taken to mean that a

neuron responds more strongly when the animal's decision is in favor of the preferred stimulus, and CP < 0.5 normally is taken to indicate a mismatch between stimulus and choice signals. However, we have shown that CPs can be artifactually biased toward values > 0.5, thus misrepresenting the true relationship between choice signals and stimulus selectivity. This problem arises when choice-related signals, presumably arising through top-down feedback, are large enough to dictate the tuning preference of the cell. In such cases, it becomes trivial that increased neuronal activity predicts choices in favor of the "preferred" stimulus (CP > 0.5).

This problem with interpretation of CPs is likely to be greater when stimulus tuning is measured during task performance. In principle, it might be avoided by determining the preferred stimulus from tuning curves measured during a task that only requires fixation, without the animal being required to make choices (e.g., Fig. 7). However, in trained animals, it is possible that choice-related signals may still contaminate measurements of stimulus tuning. This problem with CPs might also be more likely to arise for fine (vs. coarse) discrimination tasks, since the dynamic range of neural responses caused by stimulus variations may be smaller in fine tasks. Finally, CP measurements may be less likely to be biased when measured in early sensory areas where choice-related signals tend to be weaker relative to stimulus tuning. Thus, the extent to which this problem may manifest itself is likely to vary considerably from study to study, but should be examined carefully. Computing choice-conditioned tuning curves (e.g., Fig. 1) is a helpful way to examine whether this confound may be at play.

We have provided an approach that can more accurately characterize the relationship between stimulus- and choice-related response modulations, even when choice signals heavily contaminate measurements of stimulus tuning. Using this approach, we show that CPs in area VIP are generally much > 0.5, yet heading and choice signals are largely uncorrelated. Such a pattern of results may arise from top-down signals related to decision-making that are not selectively targeted to neurons based on their tuning properties. Recently, the traditional bottom-up interpretation of CPs has been questioned by multiple findings[3, 8, 22, 23] that provide strong evidence for a top-down contribution to CPs. Further research will be required to understand why top-down choice signals might selectively target neurons according to stimulus tuning in some brain areas but not others, as well as how different forms of correlated response variability may shape the relationship between stimulus and choice signals[35, 36].

Unique choice and heading signals were largely uncorrelated in VIP. However, for multisensory VIP cells (with congruent or opposite visual and vestibular heading tuning) a positive relationship might be discerned between choice and visual heading signals, irrespective of whether the choice signals relate to visual or vestibular stimuli (Fig. 5b, *left* and Fig. 6a). Hence choice signals for multisensory VIP cells seem to be aligned to visual heading signals. The opposite seems to be the case in MSTd. Namely, a positive relationship can be discerned between choice and vestibular heading signals (Fig. 2a, *right*, blue; Fig 4a, *right*; see also Supplementary Fig. 2, bottom right). For the visual condition, heading and choice signals were positively related for congruent cells, and negatively related for opposite cells (Fig. 5b, *right* and Fig. 6b, *top*). This suggests approximate directional alignment between choice signals and vestibular stimulus preferences in MSTd. This pattern of results is in line with the notion that MSTd neurons may be decoded according to the vestibular heading tuning of each neuron[6, 33]. While one could make the reverse claim for VIP (i.e., decoding according to visual heading preferences), the overall relationship between visual

heading and choice signals was weaker when looking at all neurons (Supplementary Fig. 2) and there are relatively few opposite cells in VIP. Furthermore, reversible inactivation of VIP does not impair heading discrimination[30], unlike MSTd[31], thus it may not be appropriate to interpret VIP choice signals in terms of stimulus "decoding".

Recent studies have shown that neurons in higher cortical areas simultaneously encode multiple task features ("mixed selectivity")[37–44]. Perhaps most relevant to our findings, Raposo et. al[45] have reported that parietal neurons with mixed selectivity do not demonstrate clusters of choice and cue modality signals; rather, they are distributed randomly. These results are in line with ours, however, they do not address the fundamental question of whether choice-related response modulations are predictable from a neuron's tuning to task-relevant stimuli. That is because cue modality reflects the "medium" of stimulus presentation (e.g., audio vs. visual; or vestibular vs. visual), and not the task-relevant stimulus variable (e.g., heading, in our case). We found that there is no systematic relationship between stimulus (heading) tuning and the unique (but task-related) choice signals in VIP neurons.

Interestingly, neurons in the lateral intraparietal (LIP) area, which neighbors VIP, also carry combinations of sensory and choice signals[41, 46], and similarly demonstrate large CPs[47]. LIP neurons are commonly assumed to play a role (e.g., accumulation of sensory evidence) in forming decisions[48, 49]. Yet, some evidence indicates that LIP inactivation does not affect perceptual decisions regarding direction of motion[50], similar to the observation that inactivation of VIP does not affect heading judgements[30]. This suggests that choice-related activity in parietal areas (VIP and LIP) might be driven by top-down signals that reflect the decision, rather than indicating a causal role of these areas in making decisions. However, it is also possible that choice signals in VIP reflect pre-motor response components used in some other computations. Recently Engel et al.[51] proposed that top-down decision signals underlie the learning of neuronal category representations. Decision signals that are decoupled from stimulus tuning might be beneficial for this purpose. More experimental and theoretical research along these lines will be necessary to address the many questions raised by these findings.

In conclusion, we find that VIP neurons carry strong choice signals that are largely distinct and decoupled from stimulus tuning, and thus likely reflect top-down feedback. Our findings also demonstrate that CPs need to be interpreted with care, especially when it is possible that the apparent stimulus tuning is strongly modified by choice signals. Finally, our findings raise important questions regarding the functional role of choice-related signals that are uncorrelated with stimulus tuning, spurring the need for further research on this topic.

## Methods

**Animals.** For this study, single units were recorded extracellularly from cortical areas VIP and MSTd in two awake, behaving, male rhesus monkeys (Macaca mulatta; monkeys Y and A1). In addition to examining these new data, the analyses developed in this study were also applied to old data from previous VIP[10] and MSTd[6] studies (monkeys C and U, and monkeys C and A2, respectively). Thus, in total, neuronal data from 5 monkeys were studied. The methods described here for obtaining the new data generally also pertain to the old data, apart from some specific differences described below. Please also see prior publications for further details of the methods[6, 10, 52, 53].

All procedures were approved by the Animal Studies Committee at Washington University, Saint Louis, MO (where the study began) and by the corresponding committee at Baylor College of Medicine (where the study was completed). Each animal was chronically implanted with a circular molded, lightweight plastic ring for head restraint and a scleral coil[54] for monitoring eye movements within a magnetic field (CNC Engineering, Seattle, WA). Monkeys were head fixed and seated in a primate chair that was anchored to a motion platform (6DOF2000E; Moog, East Aurora, NY). A stereoscopic projector (Mirage 2000; Christie Digital Systems, Cypress, CA) and a rear-projection screen were also mounted on the

platform. The projection screen (60 × 60 cm) was located ~30 cm in front of the eyes, subtending a visual angle of ~90° × 90°. Monkeys wore custom stereo glasses made from Wratten filters (red #29 and green #61, Kodak), which enabled rendering of the visual stimulus in three dimensions as red-green anaglyphs.

The stimulus was a single-interval linear trajectory of self-motion, headed in a primarily forward direction within the horizontal plane, with a slight deviation to the right or left of straight ahead. Self-motion was provided by vestibular cues alone (inertial motion during fixation of a blank screen), visual cues alone (optic flow simulating self-motion through a 3D star field, without inertial motion) or a synchronous combination of vestibular and visual cues. For the analyses in this study, we focus on the vestibular and visual conditions. Visual cue reliability could be varied by manipulating the motion coherence of the optic flow pattern, i.e., the percentage of dots moving coherently. The new data in this study were collected using 100% visual motion coherence, whereas coherence values for the old data are described below. The stimulus had a duration of 1 s and a total displacement of 0.13 m (resulting from a Gaussian velocity profile with a peak velocity of 0.35 m/s and a peak acceleration of 1.4 m/s$^2$).

Stimulus heading was varied in small steps around straight ahead and the stimulus set was presented using the method of constant stimuli. The monkeys' task was to discriminate, in a two-alternative forced choice, whether heading was right or left of straight ahead. During stimulus presentation, animals were required to fixate on a central target; after the stimulus terminated and the fixation point was extinguished, they reported their choice by making a saccade to one of two choice targets (located 7° to the right and left of the fixation target). A random delay period (uniformly distributed from 0.3–0.7 s) was added following stimulus offset and before the fixation point was extinguished. At the end of a trial, monkeys were rewarded for a correct heading selection with a portion of water or juice. The ambiguous condition of straight ahead was defined as 0° heading, with positive headings to the right and negative headings to the left. Stimulus heading values were ± 12°, ± 6°, ± 1.5° and 0° (for both monkeys Y and A1).

**Electrophysiology.** Single-unit activity was recorded from 307 VIP and 182 MSTd cells (new data; details regarding old data are presented below) during task performance, with both visual and vestibular stimuli. Recording locations were localized using a combination of magnetic resonance imaging scans, stereotaxic coordinates, white/gray matter transitions, and physiological response properties, as described previously[6, 10]. Most of the cells (75% of VIP neurons and 96% of MSTd neurons) were recorded using linear electrode arrays (Plexon; 16-channel U-probe, 100 μm electrode spacing). The remainder was collected using standard tungsten microelectrodes (Frederick Haer; impedance ~ 1–2 MΩ at 1 kHz). The electrode array, or single tungsten electrode, was advanced into the cortex through a transdural guide-tube using a micromanipulator (Frederick Haer) mounted on top of the head restraint ring. Data were displayed online and saved using a Plexon multichannel data acquisition system (Plexon Inc, Dallas, TX). Neurons were recorded from the target area, but were not specifically pre-screened for visual or vestibular tuning in order to reduce any selection biases. In this (new) MSTd data set, relatively few cells demonstrated robust vestibular responses (i.e., were multisensory), hence results for the vestibular condition were limited. However, these new MSTd data were supplemented by additional (old) MSTd data (described below), which had more prevalent vestibular responses, since multisensory neurons were specifically sought through a screening process. Further differences in the incidence of vestibular responses might result from individual differences between animals, as well as the fact that vestibular responses in MSTd are sometimes concentrated in posterior-medial portions of MSTd[26]. Single units were sorted offline using the Plexon Offline Sorter. A minimum of 5 full repetitions for each stimulus heading was required for inclusion in the data set. Ten repetitions were typically collected (95% of the units had 10 or more repetitions; mean = 10.5).

**Additional neuronal data.** To evaluate consistency of the findings of the correlation analyses based on the primary (new) data set described above, old data from previous studies, comprising VIP cells from Chen et al.[10] and MSTd cells from Gu et al.[6], were also analyzed. These data were recorded using single tungsten microelectrodes. Before the heading discrimination task was run, VIP neurons were pre-screened to confirm significant tuning in the horizontal plane for either visual or vestibular stimuli. MSTd neurons were also pre-screened, but included only if significant heading tuning in the horizontal plane was observed for both visual and vestibular stimuli. We used these pre-screening data to estimate the global heading tuning preferences of each neuron by calculating the vector sum of responses to 8 equally-spaced headings in the horizontal plane (45° apart): 0, 45, 90, 135, 180, 225, 270, and 315°. Responses of some cells to two additional headings ( ± 22.5), as well as non-horizontal stimuli (translations with elevation), were not used in our analyses. For the resulting 113 VIP and 183 MSTd neurons that passed the pre-screening criteria described above, the data were collected while animals performed the heading discrimination task in both the visual and vestibular conditions. Seven VIP cells were excluded here for having < 5 full repetitions for each stimulus heading, resulting in 106 VIP cells. An additional 82 MSTd cells, for which pre-screening indicated visual heading tuning only, were recorded during heading discrimination with only visual (but not vestibular) stimuli. This resulted in 265 total MSTd cells, all of which were tested with at least 10 repetitions for each stimulus heading.

For the old heading discrimination data, the stimulus had a duration of 2 s and covered a total displacement of 0.30 m (using a Gaussian velocity profile with a peak velocity of 0.45 m/s and a peak acceleration of 0.98 m/s$^2$). For the VIP recordings, heading angles were: $\pm 9°$, $\pm 3.6°$, $\pm 1.44°$, $\pm 0.58°$ and $0°$ (for both monkeys C and U). For the MSTd recordings, heading angles were: $\pm 16°$, $\pm 6.4°$, $\pm 2.56°$, $\pm 1.02°$ and $0°$ for monkey C; and $\pm 9°$, $\pm 3.47°$, $\pm 1.33°$, $\pm 0.51°$ and $0°$ for monkey A2. Visual motion coherence was set for each animal to approximately match the visual and vestibular heading thresholds, and "matching" coherence values ranged from 13-50%. For MSTd cells that were only tested in the visual condition, the data were collected for both matching and high (100%) coherence stimuli. A comparison of matching vs. high coherence data for these cells demonstrated similar heading and choice partial correlations (Supplementary Fig. 6), indicating that coherence does not substantially impact these results. For these cells, which had both matching and high coherence data, the high coherence data were used for our analyses. Behavioral data and neuronal partial correlations are also presented separately for each of the 5 monkeys included in this study (Supplementary Fig. 7).

**Data analyses.** Behavioral and neuronal data analyses were performed using custom software written for use with Matlab R2014b (The MathWorks, Natick, MA). Psychometric functions were constructed by plotting the proportion of rightward choices as a function of heading, and these data were fit with a cumulative Gaussian distribution using the psignfit toolbox for Matlab (version 2.5.6)[55]. For each experimental session, separate psychometric functions were constructed for visual and vestibular cues, and the psychophysical thresholds were defined by the standard deviations (SD, $\sigma$) of the fitted cumulative Gaussian distributions.

Neuronal heading tuning curves were constructed by computing the average firing rate (FR, in units of spikes/s) for each heading, over a time period from $t = 0.2$ s after onset of the Gaussian motion stimulus until the end of the stimulus ($t = 1$ s for the new data, and $t = 2$ s for the old data). This time period was determined by cutting off 100 ms from the beginning and the end of the stimulus epoch (where stimulus motion is close to zero) and shifting by 100 ms to approximately account for response latency. Since stimulus intensity and neuronal responses are strongest during the middle of the stimulus time-course, average FRs calculated in this manner are fairly robust to modest variations in exactly how much time is excluded at the beginning and end of the stimulus epoch. These calculations are used in analyses that require only a single average FR for each trial. Analyses that examine response metrics as a function of time relied, instead, on calculations of instantaneous FRs in a range of smaller time windows (as described below). Importantly, fine-scale heading tuning curves in this study were constructed from responses of successfully completed trials of the discrimination task, not from separate tests involving passive viewing.

Instantaneous FRs, used for partial correlation analysis over time (described below; Figs. 4, 6), were calculated as the average FR within a 0.2 s rectangular window that was stepped through the data in intervals of 0.1 s. Thus, the time index (which was taken from the center of the window) ranged from $t = 0.1$ s to $t = 1.2$ s (new data) or $t = 1.9$ s (old data). Note that, for the new data, this time range extended beyond the end of the stimulus ($t = 1.0$ s) but did not include the saccade, which could only take place after $t = 1.3$ s, due to the random delay period (0.3 s–0.7 s) that was inserted after the end of the stimulus. For the old data, a saccade could take place soon after the end of the stimulus, and hence neuronal signals were not analyzed beyond that time ($t = 2$ s).

**Partial correlation analysis.** Partial correlation analysis was performed using three parameters obtained for each trial of the heading discrimination task: the stimulus heading, the neuron's FR during the stimulus, and the monkey's choice (coded as −1 for leftward choices, and +1 for rightward choices). Two measures were then calculated: $R_{heading} = R(FR, heading|choice)$, the partial correlation between FR and heading, given the monkeys' choices (termed 'heading partial correlation'); and $R_{choice} = R(FR, choice|heading)$, the partial correlation between FR and choice, given the heading (termed 'choice partial correlation'). This analysis was performed separately for each cue type (visual and vestibular), providing 4 measures per cell: $R_{heading, visual}$, $R_{heading, vestibular}$, $R_{choice, visual}$, and $R_{choice, vestibular}$.

Partial correlations over time were calculated in the same manner, but using the IFRs (as described above, in steps of 0.1 s) as separate response measures for each time step. Since the heading and choice parameters were scalar values for a given trial, the only parameter with values that changed over time was the IFR. Thus the time-course analysis of partial correlations reflects the dynamics of neuronal activity.

**Choice probabilities.** CPs were calculated for each cell and each stimulus condition (visual and vestibular), as described previously[6, 10]. Briefly, neuronal responses were sorted into two groups based on the choices made at the end of each trial ("preferred" vs "null" choices). A preferred choice was defined as a choice in favor of each cell's preferred heading (left or right), which was determined from the heading tuning curve measured during the discrimination task, as described above. Sorting of responses by choice was done separately for each heading, as long as there were at least three rightward and three leftward choices for that stimulus. FR responses were then normalized (Z-scored), using the methodology suggested

by Kang and Maunsell[56], and pooled across headings. From these pooled responses, sorted by choice, a grand CP was computed using ROC analysis. CP > 0.5 indicates that responses were larger on trials when the animal made a choice in favor of the neuron's preferred direction for that stimulus condition.

**Data availability.** All relevant data are available from the authors.

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

## Acknowledgements

We thank Yong Gu and Aihua Chen for sharing previously recorded data, and Mandy Turner for help with animal care. This work was supported by NIH grants DC014678 and DC007620 to DEA, NIH grant EY013644 to GCD, and by The Israeli Centers of Research Excellence (I-CORE) program (Center No. 51/11) to A.Z.

## Author contributions

A.Z., G.C.D. and D.E.A.: Contributed to the design of the experiments, interpretation of results, and manuscript preparation. A.Z.: Performed the experiments and the data analyses.

## Additional information

**Competing interests:** The authors declare no competing financial interests.

