## [Peer Review File · Nature Communications]

Reviewers' Comments:

Reviewer #1 (Remarks to the Author):

This is a very nice from Deangelis's group that investigates the interpretation of choice probabilities in the context of neuronal responses in 2 motion responsive brain areas during a heading direction discrimination task. The basic point that the authors make is simple, elegant, and important: that choice probabilities can reflect both the classical interpretation of them-- in which a sensory neuron's noise is interpreted as part of the signal representation, and thus plays out as a correlation with behavioral fluctuations-- and an increasingly-supported alternate interpretation-- in which top-down signals that reflect the animal's choice are fed back to the sensory neurons, and hence also show up as a neuron-behavior correlation but with a very different meaning. Holistically, I am left pretty confident that their dissection of sensory and choice contributions to CP is valid, and the paper provides a novel means of getting at this issue that complements other studies (discussed nicely in the manuscript) that have also suggested a non-sensory / feedback / choice-related component of choice probabilities. Although the figures could use a little aesthetic modernization to improve visual clarity, the arguments and logic themselves are solid and the layered set of analyses (documented in the main and supplemental documents) appear solid and careful.

In fact, it's this last item of commendation that points to my own significant critique, which is a fairly conceptual one and admittedly might not reasonably translate into significant changes. But the meticulous partial correlation analyses (used to disentangle the sensory and choice components) strike me as an attempt to strengthen the conclusions by vetting and re-vetting the same basic analysis, when what's really needed here is an additional point of leverage. I have little doubt that the possible additions I discuss here have already been considered and I can envision many reasonable reasons why they might not be possible, available, definitive, etc. But if even one of them (or something in this spirit) proved viable and panned out, the paper's impact might be one notch more solid. Specifically, the authors themselves grapple with this issue in the discussion, explaining that one of the big problems here is that the tuning and the CPs come from the same datasets, in which the monkeys were performing a discrimination task. Thus, there's no "pure" estimate of the sensory tuning; instead it is estimated from (again, a very reasonable) dissection of the correlations.

One idea, discussed and pre-critiqued by the authors, would be to appeal to separate measurements to estimate the sensory tuning when the monkeys were not making choices. It's possible that due to the scale of the experiments these data do not exist, and sure, it's also possible that well-trained monkeys tacitly perform discrimination even during "passive" mapping. But if these data existed, even for some subset of cells, it would be worth going through. I realize that the correlation analysis couldn't be done anymore (no explicit choices to put into the analysis), but some alternate comparison of the responses to isolate a choice component might be illuminating.

Another idea would be to look at CPs computed on identical repetitions of a signal-free (0 deg) condition, to at least see that estimates of choice-related contributions are of the same magnitude and form of the grand CPs. Finally, perhaps a finer-grained analysis of the time courses might reveal correlations between the choice component and the animal's reaction time or some other novel variable of relevance.

In summary, I'm very positive about this study at both the level of rigor and conceptual importance. I'm just taking one opportunity to push for one more piece of grounding, either via reference to separate data or additional internal analyses. On that last item, the internal analyses already done (looking at congruent / incongruent cells separately) really does bolster the strength of the conclusions, and I'd suggest explaining how appealing to them strengthens the conclusions with a bit more detailed foreshadowing in the Introduction.

Reviewer #2 (Remarks to the Author):

This manuscript describes an important new finding concerning how activity in sensory neurons is related to psychophysical reports, traditionally quantified by Choice Probability. The correlation between spiking activity and choice has important implications for how we believe the brain supports behavior. They show that in area with strong choice-related signals, the traditional way of measuring CP generates artefactual results that can be misleading. Importantly, they also devise a modified method that is not subject to this confound. This is potentially an important contribution, as it makes possible the use of CP measures in more complex brain areas than simple sensory cortices.

The work is carefully done and clearly described. I have no important reservations. My only substantial comment is that a little more analysis here may allow the conclusions to be extended to previous work without requiring re-analysis. For example, it seems to me that in the original work in MT the mean CP was small enough that this effect would have had negligible impact. My reading of the paper is that they would agree with this. It might be very useful to formalize this in some way. I think that, given a selection criterion for neurons (say $d_{\text{prime}} > \text{some criterion}$, where this measure confounds choice and stimulus contributions), then partial choice correlations large enough to affect mean CP significantly would also imply a mean CP of a given size. If their simulations can be used to extract some simple rule, which allows us to say quantitatively what values of CP in existing work are safe, that would be a valuable addition.

Minor Points

Supplements top of p.3. I did not follow the argument that contributions of equal magnitude and opposite sign systematically produce CPs < 0.5 . Shouldn't they be random at this point? I.e. the regression between heading and rate should be flat on average, equally likely to follow choice or the stimulus. I understand that the simulations did behave this way, I just don't follow this explanation. Has it to do with the asymmetry in numbers – there are only two choices but more than two stimulus levels? It's the simple case with two stimulus levels where it seems clear the regression is equally likely to be positive or negative.

Line 68. “predictable from the neurons' tuning” is a slight simplification, ignoring the importance of correlations. Something like “predictable on average (over the population) from neuronal tuning” would avoid going into all the complications but still be correct.

Line 245 large... small. Might help to say “of opposite sign” in the text here, so that readers do not have to consult the figure to understand.

Supplemental figure S1D “serperate” is mis-spelled in the title.

Reviewer #1:

This is a very nice from Deangelis's group that investigates the interpretation of choice probabilities in the context of neuronal responses in 2 motion responsive brain areas during a heading direction discrimination task. The basic point that the authors make is simple, elegant, and important: that choice probabilities can reflect both the classical interpretation of them-- in which a sensory neuron's noise is interpreted as part of the signal representation, and thus plays out as a correlation with behavioral fluctuations-- and an increasingly-supported alternate interpretation-- in which top-down signals that reflect the animal's choice are fed back to the sensory neurons, and hence also show up as a neuron-behavior correlation but with a very different meaning. Holistically, I am left pretty confident that their dissection of sensory and choice contributions to CP is valid, and the paper provides a novel means of getting at this issue that complements other studies (discussed nicely in the manuscript) that have also suggested a non-sensory / feedback / choice-related component of choice probabilities. Although the figures could use a little aesthetic modernization to improve visual clarity, the arguments and logic themselves are solid and the layered set of analyses (documented in the main and supplemental documents) appear solid and careful.

We thank the reviewer for their supportive comments.

In fact, it's this last item of commendation that points to my own significant critique, which is a fairly conceptual one and admittedly might not reasonably translate into significant changes. But the meticulous partial correlation analyses (used to disentangle the sensory and choice components) strike me as an attempt to strengthen the conclusions by vetting and re-vetting the same basic analysis, when what's really needed here is an additional point of leverage. I have little doubt that the possible additions I discuss here have already been considered and I can envision many reasonable reasons why they might not be possible, available, definitive, etc. But if even one of them (or something in this spirit) proved viable and panned out, the paper's impact might be one notch more solid. Specifically, the authors themselves grapple with this issue in the discussion, explaining that one of the big problems here is that the tuning and the CPs come from the same datasets, in which the monkeys were performing a discrimination task. Thus, there's no "pure" estimate of the sensory tuning; instead it is estimated from (again, a very reasonable) dissection of the correlations.

One idea, discussed and pre-critiqued by the authors, would be to appeal to separate measurements to estimate the sensory tuning when the monkeys were not making choices. It's possible that due to the scale of the experiments these data do not exist, and sure, it's also possible that well-trained monkeys tacitly perform discrimination even during "passive" mapping. But if these data existed, even for some subset of cells, it would be worth going through. I realize that the correlation analysis couldn't be done anymore (no explicit choices to put into the analysis), but some alternate comparison of the responses to isolate a choice component might be illuminating.

We thank the reviewer for these suggestions. We did not perform separate measurements of heading tuning during passive fixation for the new datasets included in this study. However, such passive fixation data were available for

the old data sets from VIP and MSTd. We have now retrieved and analyzed these data, and the results show that heading partial correlations (measured during the heading discrimination task) are systematically related to global heading tuning preferences measured during fixation, whereas choice partial correlations are not (new Figure 7). We believe that these results do provide the additional point of leverage that the reviewer wanted. The new data strengthen our conclusions and also provide an additional validation of our partial correlation analyses. We are grateful to the reviewer for nudging us to perform these additional analyses.

Another idea would be to look at CPs computed on identical repetitions of a signal-free (0 deg) condition, to at least see that estimates of choice-related contributions are of the same magnitude and form of the grand CPs. Finally, perhaps a finer-grained analysis of the time courses might reveal correlations between the choice component and the animal's reaction time or some other novel variable of relevance.

We compared CPs calculated from the 0° heading stimulus to the grand CPs and found that they were strongly related: R (and p) values were R=0.84 ($p=1\cdot 10^{-19}$), R=0.76 ($p=1\cdot 10^{-27}$), R=0.62 ($p=3\cdot 10^{-3}$), and R=0.84 ($p=2\cdot 10^{-19}$), for VIP vestibular, VIP visual, MSTd vestibular and MSTd visual data respectively, in the new data set; and R=0.72 ($p=6\cdot 10^{-9}$), R=0.83 ($p=3\cdot 10^{-13}$), R=0.65 ($p=1\cdot 10^{-14}$), and R=0.83 ($p=2\cdot 10^{-53}$), respectively, in the old data set. The slopes of these relationships (using type-II regressions with intercept set at 0) were all between 0.94 and 1. However, we do not think that these results are an effective way to address the reviewer's main concern, and we think that the new Figure 7 has now addressed this concern satisfactorily. Thus, we have elected not to include these comparisons in the revised manuscript.

In summary, I'm very positive about this study at both the level of rigor and conceptual importance. I'm just taking one opportunity to push for one more piece of grounding, either via reference to separate data or additional internal analyses. On that last item, the internal analyses already done (looking at congruent / incongruent cells separately) really does bolster the strength of the conclusions, and I'd suggest explaining how appealing to them strengthens the conclusions with a bit more detailed foreshadowing in the Introduction.

We are confident that the new Figure 7 provides the key additional piece of grounding that the reviewer desires. We considered adding some text to the Introduction to foreshadow the dependence on congruency, but we decided against it because it would require substantial additional text to set up the issues, and we felt that this derailed the flow of the Introduction somewhat. But we suspect that this is no longer necessary with the addition of the results of Figure 7, which are now also mentioned in the Abstract.

Reviewer #2:

This manuscript describes an important new finding concerning how activity in sensory neurons is related to psychophysical reports, traditionally quantified by Choice Probability. The correlation

between spiking activity and choice has important implications for how we believe the brain supports behavior. They show that in areas with strong choice-related signals, the traditional way of measuring CP generates artefactual results that can be misleading. Importantly, they also devise a modified method that is not subject to this confound. This is potentially an important contribution, as it makes possible the use of CP measures in more complex brain areas than simple sensory cortices.

We thank the reviewer for appreciating the significance of the work.

The work is carefully done and clearly described. I have no important reservations. My only substantial comment is that a little more analysis here may allow the conclusions to be extended to previous work without requiring re-analysis. For example, it seems to me that in the original work in MT the mean CP was small enough that this effect would have had negligible impact. My reading of the paper is that they would agree with this.

We do agree with this comment, as mentioned in the Discussion. To better relate our estimates of stimulus preference from the partial correlation analysis to conventional measures of tuning, we have analyzed global heading tuning data from separate blocks of trials in which the monkeys were just passively fixating. By comparing these independent measures of stimulus preference to the heading and choice partial correlations extracted from the discrimination task data, we now provide an independent line of evidence that choice signals in VIP are dissociated from stimulus tuning (please see the new Fig. 7 and the response to Reviewer #1 above).

It might be very useful to formalize this in some way. I think that, given a selection criterion for neurons (say $d_{\text{prime}} >$ some criterion, where this measure confounds choice and stimulus contributions), then partial choice correlations large enough to affect mean CP significantly would also imply a mean CP of a given size. If their simulations can be used to extract some simple rule, which allows us to say quantitatively what values of CP in existing work are safe, that would be a valuable addition.

It would indeed be very useful if there were a simple rule of thumb regarding how large CP values can be before the confound arises. However, this cannot be deduced from the CP value alone. What matters critically is the relative balance of stimulus- and choice-related response components, such that the problem arises when the choice signals can overwhelm the true stimulus tuning. This means that the same choice signal could produce an artefactual CP in one neuron that has weak stimulus tuning and not in another neuron that has stronger stimulus tuning. Moreover, the degree of stimulus-driven modulation will depend on the details of the task and stimuli used. So there is unfortunately no threshold below or above which CPs are “good” or “bad” – one simply can't tell from the CP value alone.

We do suggest in the Discussion that a simple way to look for this potential problem in one's data is to examine choice-conditioned tuning curves.

Minor Points

1. Supplements top of p.3. I did not follow the argument that contributions of equal magnitude and opposite sign systematically produce CPs < 0.5 . Shouldn't they be random at this point? I.e. the regression between heading and rate should be flat on average, equally likely to follow choice or the stimulus. I understand that the simulations did behave this way, I just don't follow this explanation. Has it to do with the asymmetry in numbers - there are only two choices but more than two stimulus levels? It's the simple case with two stimulus levels where it seems clear the the regression is equally likely to be positive or negative.

Actually, even in the simple case of two headings, the regression will still be more strongly biased by the stimulus response. Take for example the case of two headings ($\pm h$). The FRs grouped by heading will (by definition) always comprise responses only to $+h$ or $-h$, respectively. However, in each of these groups the choices will be mixed (albeit more rightward choices for $+h$ and more leftward choices for $-h$). Thus, when heading and choice contributions are of equal magnitude but opposite sign, the regression between FR and heading will not be flat on average (this would only happen when the choices are all 100% correct). Rather, the regression used for CPs has an implicit bias to expose stimulus tuning, and average out choice tuning. Hence for heading and choice contributions of equal magnitude but opposite sign, the regression used for CPs will be characterized primarily by the heading responses (and CPs will be < 0.5). We have added this explanation to the revised Supplementary material.

2. Line 68. "predictable from the neurons' tuning" is a slight simplification, ignoring the importance of correlations. Something like "predictable on average (over the population) from neuronal tuning" would avoid going into all the complications but still be correct.

It is possible for a feed-forward model with a particular structure of noise correlation to predict that stimulus and choice signals would not be aligned for all neurons, so we appreciate the point made by the reviewer. We have softened this bit of text and modified it to make the point more generally, and we think that this addressed the reviewer's concern. The text on p. 3 now states: "In a simple feed-forward system, neuronal fluctuations should generally influence choices in a manner that is predictable from the neuron's tuning."

3. Line 245 large... small. Might help to say "of opposite sign" in the text here, so that readers do not have to consult the figure to understand.

We added this clause and agree that it makes the point clearer.

4. Supplemental figure S1D "serperate" is mis-spelled in the title.

Thank you. Corrected.

Reviewers' Comments:

Reviewer #1 (Remarks to the Author):

Wow, remarkably thorough and satisfying set of revisions. I appreciate all the work that went into additional analyses (even ones that, as the authors point out, are probably not worth adding to the manuscript itself). The current state of the manuscript leaves me with no substantive concerns or queries, and I also noted that the current version has been extensively groomed and I had trouble finding typos just to prove I'd read the whole thing carefully.

The only issue I want to mention is a request for a minor, but I think important, bit of framing/discussion-level clarification. Namely, the authors rule out the bottom-up version of choice probabilities, i.e., the VIP responses are glaringly inconsistent with simple feedforward readout. The manuscript implies that the only other alternative is feedback. But the presence of choice-correlated neural activity that's not simply feedforward sensory representation could also be a signature of the computation of the decision itself, or of a premotor response component-- both are logically possible in a feedforward framework.

Of course, the specific set of experimental results here is built on the knowledge that VIP inactivations don't have a clear effect on decisions, so the authors may in fact be able to rule out this middle-ground account. And while all the pieces of this argument are already in the manuscript, I wouldn't mind seeing it laid out more explicitly. I'll let the authors decide how best to address this.

Overall, a very interesting and important piece of work, which was nicely tightened up and polished in revision.

Reviewer #2 (Remarks to the Author):

The authors have addressed all of my concerns.

Reviewer #1:

Wow, remarkably thorough and satisfying set of revisions. I appreciate all the work that went into additional analyses (even ones that, as the authors point out, are probably not worth adding to the manuscript itself). The current state of the manuscript leaves me with no substantive concerns or queries, and I also noted that the current version has been extensively groomed and I had trouble finding typos just to prove I'd read the whole thing carefully.

The only issue I want to mention is a request for a minor, but I think important, bit of framing/discussion-level clarification. Namely, the authors rule out the bottom-up version of choice probabilities, i.e., the VIP responses are glaringly inconsistent with simple feedforward readout. The manuscript implies that the only other alternative is feedback. But the presence of choice-correlated neural activity that's not simply feedforward sensory representation could also be a signature of the computation of the decision itself, or of a premotor response component-- both are logically possible in a feedforward framework.

Of course, the specific set of experimental results here is built on the knowledge that VIP inactivations don't have a clear effect on decisions, so the authors may in fact be able to rule out this middle-ground account. And while all the pieces of this argument are already in the manuscript, I wouldn't mind seeing it laid out more explicitly. I'll let the authors decide how best to address this.

Overall, a very interesting and important piece of work, which was nicely tightened up and polished in revision.

We agree that the decoupled choice signals might reflect some premotor response component that is involved in other computations, and we have added a sentence to the Discussion (second to last paragraph of the Discussion) to acknowledge this point.

Reviewer #2:

The authors have addressed all of my concerns.

Thank you.